# User priorities for hydrological monitoring infrastructures supporting research and innovation

5 William Veness<sup>1</sup>, Alejandro Dussaillant<sup>2</sup>, Gemma Coxon<sup>3</sup>, Simon De Stercke<sup>1</sup>, Gareth H. Old<sup>2</sup>, Matthew Fry<sup>2</sup>, Jonathan G. Evans<sup>2</sup>, Wouter Buytaert<sup>1</sup>

<sup>1</sup>Department of Civil and Environmental Engineering, Imperial College London, London, SW7 2AZ, United Kingdom <sup>2</sup>UK Centre for Ecology & Hydrology, Wallingford, OX10 8BB, United Kingdom

<sup>3</sup>School of Geographical Sciences, University of Bristol, Bristol, BS8 1SS, United Kingdom

Correspondence to: William Veness (williamaveness@gmail.com)

Abstract. Observational data availability, quality, and access are major obstacles to hydrological science and innovation. To alleviate these issues, major investments are being made in hydrological monitoring infrastructures to enable data collection and sharing at unprecedented scales and resolution. These projects integrate a range of complex physical and digital components, which require careful design to prioritise the needs of end-users and optimise their value delivery. We present here the findings of multiple-methods research on end-user needs for a £38 million hydrological monitoring and research infrastructure in the UK, integrating a systematic literature review of common user-requirements with interviews of 20 national stakeholders. We find a demand for infrastructures that complement their provision of baseline hydrological datasets, where feasible, with additional services designed specifically to enable wider and more decentralised data collection. This can unlock the capacities of user communities by addressing barriers to data collection through, for example, the provision of land access, reliable benchmark datasets, equipment rental and technical support. Similarly, value can be unlocked by providing data management services, including data access, storage, quality control, processing, visualisation and communication. Our respondents further consider digital and physical spaces where users can collaborate to be critical for incubating genuine value to science and innovation. We conclude that new hydrological monitoring infrastructures require concurrent investments to build and nurture associated user, research and innovation communities, where specific enabling support is provided to facilitate collaborations. Supplementing digital and monitoring services with support for data collection and collaboration among active, value-generating user communities can produce multiplier effects from initial capital investments, by attracting longer-term contributions of ideas, methods, findings, technologies, data, training and investments from their beneficiaries.

# 1 Introduction

Many places in the world are facing unprecedented water resource management challenges from multiple pressures (Mazzucato et al., 2024., Ovink et al., 2023; Scanlon et al., 2023). For example, increasing water demand, urbanisation, ageing water systems and issues in water governance have all contributed to recent events of public controversy in the UK, where surface and groundwater pollution, water utility debts and increasing tariffs have transferred costs to the public (OFWAT, 2022; OFWAT, 2024). Climate change is also modifying global weather to increase the frequency and intensity of flood, drought and heatwave events, whilst elevating climate-risks for weather-dependent industries (Kreibich et al., 2022; IPCC, 2022; Lamb et al., 2022).

Hydrological science is struggling to address these challenges, and, despite a growing availability of remote sensing datasets, the persistent scarcity of locally-collected, shared data is still cited as a major bottleneck that holds back novel hydrological research and innovation (Chan et al., 2020; FDRI, 2022; Ovink et al., 2023; Paul et al., 2018; Buytaert et al., 2014; Sarni et al., 2018; UN-Water, 2021; Veness et al., 2025). Improving the amount, quality, resolution, coverage, range, and accessibility of hydrological datasets can therefore unlock research towards innovative solutions, whilst also supporting better decision-making in management (Nature Sustainability, 2021; Ovink et al., 2023; Paul et al., 2018; Veness et al., 2025; Vitolo et al., 2015). A growing number of global repositories such as the International Soil Moisture Network and the Global Flood Database are pooling remotely sensed and in-situ data, and have become valuable resources for advancing hydrological research (Blöschl et al., 2020; Dorigo et al., 2021; Kratzert et al., 2023). Yet these networks remain constrained by the limited availability of underlying input data, as well as integration and access barriers that limit their relevance for decision-making and research at local scales. Persistent challenges for collecting and sharing local datasets include the high costs of equipment, installation, and maintenance, as well as practical difficulties around land access, monitoring security, data management, intellectual property and data dissemination (Addor et al., 2020; Buytaert et al., 2014; Hamel et al., 2020; Paul et al., 2018; Vogl et al., 2017; Veness and Buytaert, 2025).

New technological and methodological advances are helping to address many of these challenges (Calderwood et al., 2020; Chan et al., 2020; Paul and Buytaert, 2018), with innovations in sensors, telemetry, the Internet of Things (IoT), artificial intelligence (AI), cloud computing, citizen science, and novel scientific approaches for their integration improving the potential of hydrological data systems (Paul et al., 2018; Schwab, 2017; Sarni et al., 2018; Vitolo et al., 2015; Widdicks et al., 2024). However, the development of these innovations and their uptake in hydrological monitoring and research is slow due to obstacles of limited resources, institutional capacities and technological capabilities, as well as practical challenges such as land access, data privacy agreements and intellectual property restrictions on technologies (Skinner et al., 2023; Veness, 2024; Widdicks et al., 2024).

To address these challenges, research funders are investing globally in large scale hydrological monitoring and data management infrastructures (Brantley et al., 2017). Notable projects integrating data to centrally managed digital infrastructures include critical zone observatories such as OZCAR (Critical Zone Observatories: Research and Application) in France (Braud et al., 2020; Gaillardet et al., 2018), TERENO (Terrestrial Environmental Observatories) in Germany (Kiese et al., 2018); NGWOS (Next Generation Water Observing System) in the US (Eberts et al., 2019) and federated data infrastructures in California (Cantor et al., 2021; Jensen and Refsgaard, 2018). Hydrological data infrastructures are also growing in low- and middle-income countries to the benefit of water management practitioners and hydrological researchers (Funk et al., 2019; IGRAC, 2020; UN-Water, 2021; Gale and Tindimugaya, 2019). In similar recognition, the UK government is funding a £38 million Floods and Droughts Research Infrastructure (FDRI) that will become operational in 2029 (FDRI, 2024). The primary objective of FDRI is to improve monitoring of the entire hydrological system in support of state-of-the-art research and innovation, which may be focused on floods, droughts or other practical issues in UK and international hydrology (FDRI, 2022; FDRI, 2024). Its remit and design are comparable to other national, domain-specific infrastructures, combining intensive data collection at testbed sites with sparser, strategically chosen datasets across wider networks (Nasta et al., 2025; Widdicks et al., 2024). At the same time, FDRI sits alongside broader environmental research infrastructures such as eLTER, an integrated European long-term ecosystem, critical zone, and socio-ecological research infrastructure that incorporates substantial hydrological components (Ohnemus et al., 2024), and major research partnerships such as Water4All (Water4All, 2025). FDRI is not yet embedded within these initiatives, but is being designed to remain compatible with them through shared principles, where appropriate, and flexibility for future data sharing and interoperability.

85

Hydrological monitoring and research infrastructures such as FDRI must be carefully designed to optimise long-term outcomes in research and innovation. While new projects can draw important lessons from similar international initiatives, they also need to establish user requirements specific to their national context by eliciting the perspectives of their expected users (Cantor et al., 2021; Contzen et al., 2023; Maxwell et al., 2021; Twomlow et al., 2022; Wilson et al., 2022; Nielsen-Gammon et al., 2020; Braud et al., 2020; Snow et al., 2024; Prokopy et al., 2017; Henriksen et al., 2018; Brewer et al., 2020). By clarifying where infrastructures like FDRI can generate value for their intended communities while also meeting their own scientific objectives, they can be designed to maximise impact and sustain long-term engagement (Braud et al., 2020; Cantor et al., 2021; Contzen et al., 2023; Maxwell et al., 2021; Philipp et al., 2016; Garrick et al., 2017; UN-Water, 2021; Veness & Buytaert, 2025; Zulkafli et al., 2017).

95

100

In this study, we identify end-user needs and priorities in the context of the FDRI investment. Specifically, we aim to establish what data and service needs are most important to potential users, how these shape design priorities for FDRI, and what implications they carry for the evolution of hydrological monitoring and research infrastructures more broadly. For this purpose, we deployed multiple methods, using a systematic literature review of international projects to support and cross-validate findings from interviews of 20 prospective infrastructure users. After detailing our methods, we first present the

perceived value of hydrological monitoring and research infrastructures for users, to instruct how infrastructure design can be tailored to optimise value delivery. We then present user priorities for specific fixed, mobile and digital services to deliver those benefits, and conclude by evaluating structural design priorities to ensure infrastructures deliver value sustainably.

## **105 2 Methods**

The use of multiple methods was a pragmatic choice to expand and strengthen the evidence-base informing FDRI's design (Saunders et al., 2015). A systematic review of academic literature was conducted to establish the current understanding of common user requirements from hydrological monitoring and research infrastructures (Adams et al., 2017; Haddaway et al., 2015; Page et al., 2021). The review is designed to capture learnings from projects similar to FDRI, such as other national hydrological observatories, as well as studies assessing the needs and priorities of hydrological data users for research and innovation more generally. We complement the review with semi-structured interviews of expected infrastructure users in the UK to help inform FDRI's design around the infrastructure priorities of national users (Cantor et al., 2021; Contzen et al., 2023; Maxwell et al., 2021; Twomlow et al., 2022; Wilson et al., 2022; Nielsen-Gammon et al., 2020; Braud et al., 2020; Snow et al., 2024; Prokopy et al., 2017; Henriksen et al., 2018; Brewer et al., 2020).

115


# 2.1 Systematic Literature Review

The review was guided by the PRISMA methodology (Page et al., 2021), capturing relevant studies from the Web of Science open repository and the Google Scholar database through a systematic procedure (Haddaway et al., 2015).

The search protocol ensures the presence of three elements in the search results:


1. Subject - ("flood" OR "drought" OR "hydrology" OR "hydrological") AND

This is included to capture results relevant to hydrology, flood or drought research.

2. User needs - ("information needs" OR "user needs" OR "data needs" OR "stakeholder needs" OR "user design" OR "monitoring needs" OR "stakeholder elicitation" OR "user-design" OR "user centred" OR "user centered" OR "user guided" OR "research infrastructure" OR "science infrastructure" OR "scientific infrastructure") AND

The second group of search terms ensure results include reference to hydrological data user needs or make explicit reference to a hydrological research or scientific infrastructure.

3. Monitoring/data system/research/innovation - ("monitoring" OR "observatory" OR "data" OR "research" OR "hydrometry" OR "hydrometric" OR "sensing" OR "sensors" OR "innovation" OR "innovative")

The third group of terms ensure that the studies, in their references to user needs in hydrology, make reference to user needs either from monitoring data systems or for innovation. Fig. 1 visualises the search process, the identification of relevant studies, and their subsequent screening down to the final list included in the review.

Figure 1: Procedure and results of the literature selection.



The search results of the academic and grey literature scan found no documented ex-ante (pre-implementation) user-design procedures for complete research infrastructures or hydrological observatories (Adams et al., 2017), highlighting the novelty of this study. However, there were accessible examples of ex-ante user elicitations of more limited scope, such as the design of digital platforms integrating federated hydrological datasets (5 studies). Ex-post (post-implementation) evaluations of specific hydrological research infrastructures and monitoring observatories were more common (15 studies), from which we reviewed any references to user needs and priorities for enabling research and innovation. We also included literature that is non-project specific but identifies user information needs and infrastructure priorities for supporting research and innovation in hydrology (24 studies). In this article, we integrate evidence from the review with the interview analysis.

# 2.2 Semi-Structured Interviews






In 2021, we implemented an initial set of stakeholder consultation activities, including a scoping survey (127 completed), two workshops (81 attendees), and 20 further stakeholder group discussions (FDRI, 2022). These activities yielded evidence used to inform the design of the overall architecture of FDRI (FDRI, 2022), whilst identifying FDRI's main stakeholders and the key issues to be informed through a more detailed ex-ante (pre-implementation) elicitation of their perspectives. A snowball sampling approach was used to contact potential respondents, which benefitted from FDRI's network of key informants covering the science, industry, and civil society sectors (Gumucio et al., 2021; Saldana, 2021). The sampling was focussed as interviews progressed to represent the key expected organisational sectors of end-user, as identified during the prior consultation activities, which notably informed the need to sample a range of academics to cover different research areas (Fig. 2; Saldana, 2021). Within these groups, we specifically targeted individuals recognised by peers as knowledgeable about hydrological data systems and research infrastructures, with all respondents holding at least five years of relevant experience in their sector. Sampling experienced participants allowed for more substantive reflections on design elements and priorities. Interviews of 20 participants took place between November 2023 and March 2024, with sampling continuing until major organisational sectors were represented and where the amount of new information arising in the interviews was low (Saldana, 2021). The FDRI project intends to continue the interviews at more local scales and with more targeted questioning as the infrastructure design becomes more detailed. As such, these interview perspectives represent a first pass of user-priorities, upon which future elicitations and FDRI's corresponding local infrastructure design can be adapted. The participants have been pseudo-anonymised with labels representing their organisational sector and no further identifying information (Fig. 2).

Figure 2: Organisational sectors of the respondents. The letter in brackets is used to reference the pseudo-anonymised respondents in the analysis.

A semi-structured interview approach ensured a consistent structure that addressed key questions, whilst leaving space for emergent information unfamiliar to the interviewer to be pursued through follow-up questioning (Galletta et al., 2012;

Mojtahed et al., 2014). The interview template was informed by prior stakeholder consultations and iterative design within the FDRI team to ensure that questions reflected priority areas for user-input. The questions covered a range of topics, many of which targeted more detailed components of FDRI's operational design, such as training activities, the identification of existing partnerships and the scoping of long-term funding opportunities (FDRI, 2022; FDRI, 2024). We present results from analysis of a sub-set of those questions, listed below, which more fundamentally interrogated the potential value of the infrastructure for research and innovation and how to optimise that value through a user-responsive design.

# 185 Organisational background






- Which organisation(s) are you affiliated with?
- What is/are your role(s) in that/those organisation(s)?
- How would you classify your organisation(s)?

# Perceived value

• What do you see as the value of the FDRI programme with respect to innovation? Why?

# Infrastructure Priorities

- From your perspective, what modern technologies would you like to see collecting data, and what specific functionality is required in terms of fixed infrastructure (operated by FDRI)? Why?
  - ... in terms of mobile infrastructure? (available for community use):
  - From your perspective, what digital infrastructure would you like to use?
  - What types of 'social' innovation would you like to see? Why?

# Barriers to innovative data collection and additional services

- What are the current barriers to field testing of innovative technologies?
  - What other services would you (or your organisation) like from these testbed sites? Why?
  - As a member of the community using FDRI interested in its continued technological innovation, what types of exchange would you like to see? Why?
- The qualitative interview responses were recorded manually into a secure webform by the interviewer during and following the completion of each interview. The database of responses was then analysed through qualitative coding of the responses and thematic analysis (Creswell, 2009; Saldana, 2021). This analysis approach enables quantifications of frequent responses among the different stakeholder groups, whilst also ensuring a structured and unbiased approach to interpreting the key qualitative findings and recommendations from the user consultation (Patton, 2014; Saldana, 2021). The qualitative coding

used an inductive approach for all questions, whereby codes and themes are not pre-set in advance, but instead emerge from the data through the analyst's interpretation of participant responses (Saldana, 2021). Each question was analysed separately, with interpretive codes assigned to objective-relevant information within each answer. As the analysis progressed, the repeated occurrence of certain codes and the interpretation of relationships between them enabled their organisation into emergent themes and sub-themes. To standardise these emergent codes and themes from the analysis, three sequential rounds of coding were completed (Galletta, 2013; Saldana, 2021).

During the initial rounds of coding, we observed that many interviewee suggestions were not limited to the specific questions asked but also converged around three broad areas of emphasis: services enabling data collection, services enabling community research and innovation, and a need for adaptive infrastructure design. To capture these cross-cutting recommendations, we additionally organised relevant recommendations from across all interview questions into these abductive thematic coding groups (Saldana, 2021; Saunders et al., 2015). The findings are presented in Sect. 3.3 as "structural design priorities." Although more interpretive than the inductive results in Sects. 3.1 and 3.2, they belong in the Results because they reflect emergent and recurrent points raised independently across the questions, supported by literature evidence, and are directly relevant to the infrastructure design (Galletta, 2013; Saldana, 2021; Saunders et al., 2015). Following completion of the thematic analysis, data visualisation in Figures and Tables, and draft of an academic manuscript, the draft was shared with 4 senior members of FDRI's project team for feedback. Given their relevant expertise and prior experience on the project, this process provided validation that the study interpretations and conclusions were not significantly contrary to their interpretations, whilst ensuring findings were also effectively communicated.




The results of the thematic analysis are presented in quantitative thematic plots, including simple tables and variable symbol diagrams to represent the number of participants referenced by each primary code (Galletta, 2013; Saldana, 2021). The more qualitative elements of the findings are presented and integrated with those from the systematic literature review though narrative analysis and direct quotations (Saldana, 2021; Mills et al., 2006; Creswell, 2009). In the analysis, references to evidence from the systematic review use standard Harvard referencing, whilst information referenced to interview respondents are represented in square brackets containing their organisational code and a unique number (Fig. 2). Finally, we present a conceptual model within the discussion (Sect. 4.1), which is an interpretive visualisation designed by the authors of this article and validated through feedback from the wider FDRI team to communicate key findings from the multiple-methods analysis (Mills et al., 2006; Patton, 2014).

## 3 Results

#### 255 3.1 Value Proposition for Research and Innovation

The first part of our analysis considers the value proposition of FDRI from the perspective of prospective users. Interviewees articulated four recurring themes of expected value: user community networks, data quantity & quality, testing spaces, and access to innovations (Table 1). In the analysis, we also draw on the literature review and indicate where comparable themes have been discussed in other international studies.


Table 1: Thematic summary of user perceptions of FDRI's potential added value for research and innovation in UK hydrology [Q27: What do you see as the value of the FDRI programme with respect to innovation? Why?]. The number of participant responses for each code is indicated in brackets.

| Value Theme             | Sub-theme                     | Code (frequency)                               |     |
|-------------------------|-------------------------------|------------------------------------------------|-----|
| user community networks | research & innovation network | development of user/innovation community       | (6) |
|                         |                               | communication with wider community             | (4) |
|                         |                               | academia-industry connections                  | (1) |
|                         | collaborative projects        | collaborations                                 | (7) |
|                         | coordination                  | stakeholder (long-term) coordination           | (2) |
|                         |                               | learnings for practitioners                    | (1) |
|                         |                               | developing previous work further               | (1) |
|                         |                               | data storage                                   | (1) |
|                         |                               | data sharing                                   | (1) |
| data quantity & quality | quality baseline monitoring   | reliable/long-term benchmarks for testing      | (6) |
|                         |                               | improved quality of measurements               | (3) |
|                         | interoperability of data      | correlating between datasets                   | (1) |
|                         |                               | data linking to models                         | (1) |
|                         |                               | integration of data                            | (1) |
|                         |                               | catchment approach                             | (1) |
|                         | scale                         | scale                                          | (2) |
|                         |                               | access to wider range of data                  | (1) |
| testing spaces          | technology testing            | experimental space for innovative technology   | (6) |
|                         |                               | validating and creating business case for tech | (1) |
|                         |                               | solution-oriented innovations                  | (1) |
|                         |                               | reduced barriers to site testing               | (1) |
|                         | method testing                | experimental space for innovative methods      | (4) |
|                         |                               | portal approach                                | (1) |
| access to innovations   |                               | wider access to innovative equipment           | (3) |
|                         |                               | diversity of innovation                        | (2) |



The modally identified value theme of *user community networks* contrasts with traditional perceptions of monitoring infrastructures as largely generating their value to research and innovation through the datasets they provide. Instead, our respondents emphasise the value generated by creating and engaging in a community of monitoring infrastructure users and contributors. Respondents highlighted that, by creating a focal point to draw together stakeholders from different industries and research backgrounds, monitoring infrastructures can foster innovation when collaborations form among users with unique combinations of expertise [A1, A4, A6, A8, A9, S1, S2]. This emphasis on cross-sector collaboration is echoed in international experiences, where data infrastructures have been shown to support innovation by convening diverse communities of practice

(Baron et al., 2017; Peek et al., 2020; Fleming et al., 2024; Holzer et al., 2019; Roy et al., 2020; Sartorius et al., 2024; Harrison et al., 2024; Averyt et al., 2018; Widdicks et al., 2024). Two start-up representatives highlight that these combinations can generate novel approaches that capitalise upon respective partner strengths to identify and address inter-disciplinary knowledge gaps [S1, S2] (Peek et al., 2020). Four interviewees suggested that partners in such collaborative projects address weaknesses by filling expertise gaps and cross-validating methods and results [A6, A2, S1, A4] (Averyt et al., 2018).






"bringing in different opinions and ideas from different places is how to truly innovate"

[S1]

The value of community collaboration is increasingly recognised by data infrastructure providers internationally (Baron et al., 2017; Peek et al., 2020; Fleming et al., 2024; Holzer et al., 2019; Roy et al., 2020; Sartorius et al., 2024; Harrison et al., 2024; Averyt et al., 2018; Widdick et al., 2024), as reflected by trends towards investments aiming to facilitate 'convergence' and 'synthesis' research, supporting collaborations among stakeholders and researchers from different backgrounds (Fleming et al., 2024; Peek et al., 2020; Baron et al., 2017). Eight respondents recommended that FDRI set aside resources to sustain a community integrating data users, providers, and major stakeholders in research, innovation, and water resources management [T1, A2, A4, R1, A6, A10, C2, I1]. The importance of investing in such community-building has also been demonstrated in other infrastructure contexts, where sustained user engagement is critical to long-term scientific and operational impact (Holzer et al., 2019; Prokopy et al., 2017; Sartorius et al., 2024; Gaillardet et al., 2018; Cantor et al., 2021; Henriksen et al., 2018; Peek et al., 2020; Harrison et al., 2024; Tate et al., 2021; Kiese et al., 2018; Widdicks et al., 2024). A stakeholder elicitation for an integrated hydrological data system in California concludes that this community creation is critical even to the sustainability and long-term operation of the monitoring system beyond its initial capital investment (Cantor et al., 2021; Harrison et al., 2024):

"Ensuring that an environmental data system is sufficient, accessible, useful and used hinges on meaningful, ongoing relationships with data users"

- (from Cantor et al., 2021)

In the second theme, respondents identify the evident value of *data quantity and quality* for state-of-the-art research and innovation. Six interview respondents particularly highlight the value of open access to high-quality, long-term baseline monitoring [A1, A2, A3, T1, A5, N1] (Cantor et al., 2021; Widdicks et al., 2024). Co-locating a large range of hydrological parameters at high resolution enables interrogation of novel research questions enabled by unprecedented levels of data access and complementarity [A1, T1, N1] (FDRI, 2022). The presence of long-term benchmark datasets also creates ideal *testing spaces* for the deployment and validation of innovative methods, models, and technologies, which respondents believe can catalyse their development [A2, A3, A4, S1, S2, A7]. Three respondents suggested that, if *access to innovations* of hardware, software or methods can then be shared within enabled user communities and innovation spaces, synergistic value should be generated for researchers, innovators and other monitoring infrastructure users [A4, A7, S2]. Connected communities can

share innovations [S2], jointly address mutual challenges such as land access or telemetry [A4, A7], and their collective research and innovation outputs can generate publicity, new partnerships and opportunities for funding [A2, A9, S1] (Widdicks et al., 2024).

## 3.2 Monitoring and Digital Service Priorities



Next, we identified the specific digital and monitoring products and services that prospective users identify as priorities to deliver on expected themes of value (Table 2). Because user elicitations are typically iterative across space and time, we treat these findings as a first cross-sectional input to national infrastructure design to be refined in subsequent rounds (Braud et al., 2020; Cantor et al., 2022; Widdicks et al., 2024). Interview respondents notably discussed whether monitoring should be provided by the infrastructure or collected by FDRI's user community with enabling support. We analyse this discussion point further as a key structural design principle in Sect. 3.3.

Table 2: Thematic summary of desired digital and monitoring products and services within FDRI (Q11, Q12, Q14).

Included codes refers to labels designated to participant responses during thematic analysis. 'frequency' represents the number of times a code within the theme was allocated to a response.

| Infrastructure Components   | Theme (frequency)                | Included Codes                                                                                                                                                             |
|-----------------------------|----------------------------------|----------------------------------------------------------------------------------------------------------------------------------------------------------------------------|
| digital components          | accessibility (13)               | APIs (Application Programming Interfaces), data platform, real-time data access                                                                                            |
|                             | processing & visualisation (9)   | data visualisations, easy to use data formats, data processed to target audience interests, community-friendly platforms, processing tools                                 |
|                             | interoperability (7)             | integration with other data platforms, avoid 'reinventing wheels', interoperable data                                                                                      |
|                             | quality assurance/control (6)    | quality assurance/control, data standardisation                                                                                                                            |
|                             | transmission (4)                 | transmission support in remote locations                                                                                                                                   |
|                             | collaboration infrastructure (4) | academic code publishing repository, open science, reproducibility procedures, digital community for collaborations                                                        |
|                             | storage (3)                      | secure data storage                                                                                                                                                        |
|                             | support services (3)             | backend support, Q&A (Question & Answer)                                                                                                                                   |
| fixed monitoring components | water quality (24)               | surface water quality, turbidity, nutrients, electrical conductivity, total dissolved solids, pH, isotopic tracers, nitrates, phosphates, eutrophication, dissolved oxygen |
|                             | channel parameters (22)          | surface water level, velocity, discharge, flow, sediment transport                                                                                                         |
|                             | surface extent (7)               | floodplain water monitoring, live imagery, wetland extent, reservoir flow                                                                                                  |
|                             | groundwater (4)                  | groundwater level, groundwater quality                                                                                                                                     |
|                             | biological (3)                   | beaver channels, biosensing tech, biological productivity                                                                                                                  |
|                             | technical (3)                    | Internet of Things (sensor agnostic) units, fixed drone passes, transmission infrastructure                                                                                |
|                             | atmospheric (3)                  | precipitation, evaporation                                                                                                                                                 |
|                             | soil (2)                         | soil moisture                                                                                                                                                              |

|                   | marine (1)           | marine buoys                                                                     |
|-------------------|----------------------|----------------------------------------------------------------------------------|
|                   | other (4)            | satellite lidar, remote sensing data, health & safety,                           |
|                   |                      | location data, historic data                                                     |
| mobile monitoring | multi-parameter (28) | UAVs (Unmanned Aerial Vehicles), ARC-boats, floating                             |
| components        |                      | sensors, pole mounted sensors, citizen data collection                           |
|                   | flow & velocity (14) | ADCP (Acoustic Doppler Current Profiler), image                                  |
|                   |                      | velocimetry, flow meters, bathymetry, lidar platforms                            |
|                   | flood extent (7)     | flood extent, drones after events                                                |
|                   | water quality (6)    | high-resolution water quality data                                               |
|                   | biological (3)       | metabolism gas chambers, throughfall, stemflow, nature based solution evaluation |
|                   | atmospheric (1)      | rain gauges                                                                      |
|                   | other (2)            | sediment transport, CRNS (Cosmic-Ray Neutron Sensor)                             |
|                   |                      |                                                                                  |

## 3.2.1 Monitoring infrastructure products and services



- In FDRI, the monitoring infrastructure is conceptualised in terms of fixed and mobile components. The former consists of instruments such as flow gauging and weather stations that remain on site for long periods of time, and potentially the entire lifespan of the infrastructure. Mobile components do not have a fixed location but are instead used for flexible, short-term monitoring, which may range from individual events to short campaigns.
- For *fixed components*, the level of perceived importance varies according to specific stakeholder interests. For example, demand for river channel and water quality measurements is more common among those with flood research interests, compared to groundwater and soil moisture measurements for those involved in drought and agricultural research. Despite a large variance in recommended parameters, the co-location of complimentary parameters within high monitoring intensity testbed catchments is commonly considered a priority for innovative research [A1, A2, A3, T1, A5, N1].

Mobile components can both be deployed by FDRI operational staff, but also made available for hire by infrastructure users. A specific use case flagged by four respondents is for upper reaches of catchments, where high-grade fixed instruments on small tributaries might be less cost-effective [A1, A2, A8, C1]. Interviewees also recommended mobile deployments for short-term events such as floods or pollution incidents, and they proposed that digital services could include notification and coordination features to prompt intensified data collection by users, technicians, innovators and citizen scientists during or after events [A2, A8, C1]. A wide range of relevant equipment is also flagged, including multi-parameter Unmanned Aerial Vehicles (UAVs), floating sensors and handheld probes, all of which can offer periodic surveys with similar parameters to those collected at fixed instrument sites but at higher spatiotemporal resolution (Table 2).

Lastly, we also identify strong support among respondents to include expanding social innovations, such as citizen science and community co-design [S2, C1, C2, T1, A5, R1, A7, N1, A9, N2, S1]. Citizen scientists explain that integration of their existing

projects would be a cost-effective opportunity to tap into motivated, experienced and locally knowledgeable groups, expanding the monitoring and research capacity of the infrastructure's engaged user community for mutual benefit [C1, C2].

# 350 3.2.2 Digital products and services

The modal recommendation from the interviews for digital services is a platform that aggregates data from different sources and locations [T1, A5, A9, A10, C2]. Interviewees emphasised that such a platform should be openly accessible and, where feasible, provide near real-time and visualised data that is navigable by the public, while remaining useful for expert users via Application Programming Interfaces and download options [S1, S2, I1, A10] (Dallo and Marti, 2021; Jones et al., 2015). Cantor et al. (2021) and Widdicks et al. (2024) recommend polycentric (federated) approaches to building such a platform. Instead of building a single monolithic platform, a combination of linked and interoperable platforms may be more flexible and cost-effective; for example, by supporting the integration of more localised activities or specific projects (Cantor et al., 2021; Widdicks et al., 2024). At the same time, to avoid dispersion and lack of integration, a fully data-aggregating platform is recommended by Cantor et al. (2021) to improve data discoverability, ease of access and state-level user engagement. As the platform should aim to integrate data contributions from a range of sources, respondents highlight the need for adaptable data sharing agreements and accommodation of intellectual property interests [A5, A7, N1, S1, S2]. To increase the range of data available, interviewees recommend that infrastructure providers seek secure data sharing agreements with other existing infrastructures [A3] (e.g. population censuses, disaster risk monitoring and remote sensing platforms), where the datasets are transferrable [T1, N1, A9, C2] standardised [A6, T1] and inter-operable [T1, N1, A9] (Dahlhaus et al., 2015).

In an enabling infrastructure, it is to be expected that a substantial proportion of the data will be contributed by users. As such, prospective users and recent literature both emphasise needs for transparency over data origins, processing history and prior quality control procedures (Table 2; Fileni et al., 2023). Digital Object Identifiers (DOIs), reproducibility repositories and metadata uploads are suggested as ways of achieving this [T1, A9] (Braud et al., 2020; Cantor et al., 2021), with the associated recognition and opportunities for data providers providing additional incentives for continued contributions. The FAIR (Findable, Accessible, Interoperable, Reusable) principles are emphasised in recent literature as suitable requirements for data inclusion (Braud et al., 2020; Cantor et al., 2021; Widdicks et al., 2024; Wilson et al., 2022), as well as the standards of the Open Geospatial Consortium for remote sensing and vector data (Kmoch et al., 2016). The detailed implementation of these standards within FDRI will be defined in later design stages through engagement with equivalent infrastructures and testing with early adopters. Specific functionalities suggested by respondents to support user-driven data production include secure cloud storage for datasets, ideally at low or no cost [A2, A4, A5], as well as backend support [A3, A4], technical assistance [A9] and support with data standardisation [A6, T1], all of which prospective users consider would incentivise and facilitate data contributions (as further elaborated in Sect. 3.3.1). For integrating external data contributions, respondents also emphasised the importance of harmonising measurement protocols to ensure comparability across sites and contributors [T1,

A6, A9], for which the eLTER research infrastructure recently defined an adoptable Framework of Standard Observations (Zacharias et al., 2025).

Four respondents with backgrounds studying or actively managing hydrological hazards explain the benefit of data availability in real-time to inform public awareness and active disaster risk management decisions [A7, A10, C2, I1]. Specific approaches that can support this function include: automated data quality control that is manually verified following anomaly alerts and during periodic audits [R1, A8, N2], visualisation in a geographical information system context [A2, A9, A10] and stakeholder alerting for data extremes [A2, C1, S1] (Braud et al., 2020; Kmoch et al., 2016; Dallo and Marti, 2021). Elicited user-groups in Nordic states also emphasise the benefits of linking digital platforms to social media sites for real-time data dissemination and public engagement (Henriksen et al., 2018). These platforms, particularly X (formerly known as Twitter) and Facebook (Stephenson et al., 2018), are used regularly by researchers and practitioners as well as the public, and they are an under-utilised medium for communication, awareness-raising and co-ordination [N2] (Stephenson et al., 2018). Any such use of data on these platforms would need to comply with data-sharing agreements and personal data protection requirements.

Despite these potential benefits, two potential users caution that providing real-time data access can create operational reliance on the data, with high expectations of platform uptime and performance [C2, N1]. Two academics warn that this may also go against the core mission of infrastructures like FDRI if they are primarily intended to support research and innovation rather than replacing operational infrastructure such as flood information systems [A7, A9]. Investing in ultra-reliable real-time services for operational systems may divert resources from core research and innovation functions that rely less on immediate data accessibility [A7, A9]. Nonetheless, there are many opportunities for aggregated monitoring infrastructures to provide new insights, validation and other data services for operational systems [A2, A7]. Furthermore, real-time data can support novel practical research applications such as rapid post-event studies and citizen science campaigns, whilst providing additional incentives for user contributions if data can be immediately viewed. Hence, fulfilling these opportunities whilst managing expectations and averting misuse in risk contexts requires planning and potential partnership with other data services acting in the public interest (Collins et al., 2016; Dallo and Marti, 2021; Stephenson et al., 2018).

#### 3.3 Structural Design Priorities for Value Delivery







Cross-cutting themes emerged from the interviews that extend beyond question-specific findings, supported by evidence from the literature review. These emphasised the need for hydrological monitoring infrastructures to move beyond traditional models where providers act mainly as data collectors, proprietors and distributors, towards designs that actively engage and support their user communities. Respondents and literature alike highlighted that such engagement can expand data availability, strengthen research and innovation outcomes, and improve long-term sustainability (Widdicks et al., 2024; Cantor et al., 2021). From this analysis, we identify three structural design priorities for hydrological data infrastructures, which are examined in the subsections that follow.

Firstly, our respondents emphasise that monitoring infrastructure requirements are local-context specific, influenced by, for example, pertinent issues in the local catchment, local climates, pre-existing stakeholder activities and local capacities [C1, C2, A7, A3, A5, A6, A7, A8, A9, N1]. As such, they recommend iterative, finer-scale user elicitations during their rollout to adapt the infrastructure design to local requirements. The recommendations from local user elicitations should be reviewed alongside the preferences of non-local researchers, who may prefer alternative monitoring or support arrangements towards more generalisable research themes. In such cases, having infrastructure-facilitated spaces for discussion (such as workshops and online forums) can discover and prioritise areas of mutual interests, as well as areas where suitable compromise is required in infrastructure design [S1, C2, A3, A7, A8]. Periodic evaluations should be continued indefinitely to respond to dynamic user needs and set up long-term "adaptive management cycles" (Braud et al., 2020; Cantor et al., 2022; Widdicks et al., 2024).




Second, respondents widely made recommendations to complement the provision of core datasets with additional services that 425 are enabling of data collection where possible, through a suite of data collection support for its community of users and contributors [T1, A9, I2, A1, A2, A3, A4, R1, A7, S2]. An ex-ante elicitation of Nordic stakeholders for a web-based flood management tool reached a similar finding that, by supporting monitoring among an infrastructure's entire user community, data collection capacity can be expanded far beyond that of the central institution with its internal funding capacities alone (Henriksen et al., 2018; Kruczkiewicz et al., 2021). Respondents believe that community-led monitoring is also more likely 430 than centrally-led monitoring to address relevant data gaps according to the dynamic data needs of local infrastructure user communities [C1, C2, A3, A5, A7], which is a view shared by recent studies (Kiese et al., 2018; Harrison et al., 2024; Widdicks et al., 2024). However, three respondents and the authors of this study emphasise that investments supporting data collection are contingent on having sufficient monitoring capacities, motivation and incentives to participate among stakeholders in each hydrological catchment [C1, R1, A6]. We also suggest that infrastructure providers consider whether expenditures on these 435 enabling services will have opportunity costs, such as reducing the coverage of their provided datasets, when deciding how to allocate resources. Therefore, the extent to which monitoring responsibilities can be decentralised is context-dependent and in many cases the transition may be a gradual process, where infrastructure providers are expected to "take the lead" through demonstrative priority monitoring installations [R1, A8, S1, A5, A6, A7] that deliver local value and deepen user community engagement, while they work to gradually develop data collection capacities and incentives among local infrastructure users 440 [A5, A6, A9, C1, C2, N1, I1]. Recommendations of how infrastructures can provide data collection enabling support, principally by addressing the barriers to field data collection, are outlined in Sect. 3.3.1.

Third, in line with the expected value generated by the creation of an active infrastructure user community (Table 1), there is a clear recommendation for active support that enables networking, sharing and collaborations to catalyse research and innovation among users. Recommendations for specific support enabling collaboration and innovation are analysed in Sect. 3.3.2.

## 3.3.1 Services Enabling Data Collection




Participants perceive a range of barriers to field implementation of monitoring innovations (Fig. 3) and recommend enabling infrastructure services that address them.

Figure 3: Thematic summary of perceived barriers to field testing of innovations in response to Q21: We will be using sites as innovation testbeds... What are the current barriers to field testing of innovative technologies? The symbols scale to the number of interview references made to each code (light blue) and theme (dark blue), and dashed lines represent overlap between themes.

Access is the modally perceived barrier to field testing innovations. Whilst distance [A4, S1] and a lack of safe physical access [I2, A4, R1] are an access barrier at some monitoring sites, respondents refer principally to the challenge of securing land and monitoring permissions [A3, A4, A6, I2, S1, C1, C2]. A recommended priority for supporting services, therefore, is to engage landowners, regulators, ethics committees and environmental authorities to ensure a simpler process for securing safe access and monitoring permissions for a wide variety of users at testbed sites [A3, A4, A6, I2, S1, C1, C2]. Such engagements are expected to address *local support* and *physical* barriers, by formalising interactions between infrastructure users and local stakeholders to ensure long-term support for data collection at recognised physical access points [C1, S1, R1]. This may also reduce the risks of sensor damage or theft commonly experienced at experimental sites [A6, A8]. For FDRI, prospective users recommend high-accessibility testbed catchments to function as exemplars of high intensity monitoring, which can host novel research projects and dedicated spaces for innovation testing [A2, S1, S2, A5, N2] (FDRI, 2022; FDRI, 2024; Wagenbrenner et al., 2021). Beyond testbed sites, there is also demand among respondents for procedures to support land access nationally.

where the infrastructure acts as a broker and facilitator between researchers and third parties responsible for access permissions [A4, N2, S1, C1, R1, A7, A8].

Interviewees also state a need for a range of *supporting infrastructure* for their implementation of monitoring technologies. Chosen sites for co-located monitoring should provide power [A8, S1], robust telemetry solutions through 2-5G or LoRa (long-range) networks [I2, S1], and a long-term installation of commercially approved sensors to ensure comparable benchmark datasets are available for technology and data validations [R1, A6, A7]. They also recommend an availability of support technicians in the infrastructure to offer technical support, installation services, and the rapid troubleshooting of issues [A5, A6]. An employed technician can take further responsibilities in coordinating the sharing or renting of monitoring technologies between members of the user community [A6].

The provision of supporting infrastructure services and access arrangements should additionally alleviate *time* and *cost* barriers, by reducing the time and money spent visiting monitoring installations and resolving minor technical problems [I1, A4, S1, N2]. This can free up partner resources to address the *sensors* barrier through better testing and development, as issues of reliability and robustness remain a concern for automated data collection [A1, A8]. An enabling infrastructure can facilitate the sharing of helpful resources to this end, such as open-source code, training, and opportunities for gaining technology investments [A6, S2, S1].

Users further recommend breaking norms of a one-directional flow of information from data producer to data user, by supporting social innovations for data collection. Citizen science is recommended by ten respondents to improve data coverage, data validation, community engagement and subsequent value creation [T1, R1, A5, A6, A7, A8, N2, S1, S2, C1] (Buytaert et al., 2014; Paul et al., 2018). Existing hydrological citizen science projects within infrastructure catchments are considered significant opportunities to cost-effectively catalyse data collection efforts, by providing financial, operational or other desired support in exchange for data, research participation and other practical actions [T1, R1, A6, A7]. A wider range of social innovations beyond citizen science also features strongly in the interviews, such as participatory monitoring, co-design and opportunistic data collection, to further improve datasets and associated co-benefits [A8, A10, S1]. For FDRI, an innovation co-ordinator is recommended by the regulator representative to organise the integration of social innovations into the monitoring infrastructure and its community [R1].

## 3.3.2 Services Enabling Community Research & Innovation







Creating and sustaining an active community of users, contributors and innovators requires investment into the creation of digital and physical spaces for inter-engagements, which is recommended as an additional service by thirteen respondents [T1, A2, A3, A4, A5, R1, A6, A7, A8, N2, S1, S2, A10] (Baron et al., 2017). For FDRI, informants recommend innovation events to showcase innovations [T1, A6], webinars and seminars for regular user engagement and marketing of FDRI activities to

potential partners [A2, N2]. Unified digital collaboration spaces can be integrated with data platform(s), which can host spaces for forum, Q&A, data sharing, community communications, event organisation, research coordination, and collaboration opportunities [A2, A7, A9, A10]. Newsletters or equivalent communications are recommended to keep user communities informed with current activities, research and opportunities [C1, T1, A1, A3, A5, A7, A2]. Small businesses suggest avoiding monopolisation of engagement by larger companies, stating that genuine innovation happens when small-scale innovators from different backgrounds and areas of expertise are given enabled spaces to exchange ideas, collaborate and create in intellectual property (IP) secure spaces [S1, S2]. Creating a network of start-ups, innovation incubators and investors can create vibrant digital and in-person spaces for private sector innovation [S1, I1]. Concerns over intellectual property, specifically regarding technology and data sharing, can be addressed directly by the development of adaptable template agreements [I2, S1, S2, C1].

Beyond the creation of enabled collaboration spaces, institutions providing hydrological monitoring infrastructures can actively catalyse innovative collaborations. For example, the CONVERGE project in the United States of America actively coordinates its research community by defining research priorities, facilitating partnerships, and providing updates that increase awareness of active research, share (honest) methods and findings, and avoid research activity redundancies (Peek et al., 2020). The direction of any coordination can be guided by workshops with involved stakeholders, where respective goals and an overarching research and innovation strategy is agreed (Fleming et al., 2024; Holzer et al., 2019). Training programmes are considered critical among respondents for ensuring that potential users have the capacity to engage with the monitoring infrastructure [A2, A4, A5, R1, A6, A8, N2, I1, A10, S1]. Training also increases stakeholder awareness and understanding of other related disciplines of research, which helps infrastructure users to consider potential collaborations with other disciplines [S1] (Peek et al., 2020; Harrison et al., 2024; Kiese et al., 2018). Experiences from the TERENO observatory in Germany additionally show the benefits of joint measurement campaigns as another space for catalysing cross-disciplinary research and collaboration (Kiese et al., 2018).

4 Discussion







# 4.1 Conceptual Design of a User-Enabling Monitoring and Research Infrastructure

Our results indicate that adding services *enabling* data collection and community innovation can substantially increase engagement, contributions, and the longer-term impact of hydrological monitoring and research infrastructures such as FDRI. In Fig. 4, we conceptualise this effect through a model visualising a user-enabling hydrological monitoring and research infrastructure.

Figure 4: Interpretive conceptual model summarising recommendations for a user-enabling hydrological monitoring and research infrastructure. The central Venn diagram reflects user-recommended design priorities for services enabling data collection (Sect. 3.3.1) and services enabling community research and innovation (Sect. 3.3.2) The respective inputs and output value for the infrastructure provider and user community are also shown (as informed by Table 2).



In the model, the infrastructure provider's inputs of funding, coordination and operational resources sets up a range of services to catalyse data collection (as in Sect. 3.3.1) and research and innovation (Sect. 3.3.2) among the infrastructure's user community. We show these integrated digital, monitoring and support services within the Venn diagram (summarised from Sect. 3.2 and Sect. 3.3), which deliver value towards the community members' objectives (as defined in Sect. 3.1). Benefits from these services incentivise a range of return inputs considered by other studies to be critical to the infrastructure's long-term sustainability (Cantor et al., 2021; Peek et al., 2020; Gaillardet et al., 2018; Holzer et al., 2019; Harrison et al., 2024; Widdicks et al., 2024). These include contributions of data and equipment by the user community to expand the monitoring network, as well as new results, methods and technologies from associated research and development activities. Over time, some users will have a willingness to pay for appropriate services such as data storage, telemetry or data analytics to support

the infrastructure's cost recovery. Evidence of value will also attract additional finance options, such as research grants, public funding, private industry contributions, private equity for innovations, and options for debt finance if revenues approach or exceed operational expenditures. Collectively, this is expected to support a sustainable financial model for continuing long-term operation, which may be a combination of public and private funding, supported by revenues from paid services.

These priority areas reflect a growing demand for monitoring infrastructures that better enable two-way engagement with their user communities. This demand for 'enabling' support and two-way exchange reflects the improving capacities of decentralised hydrological stakeholders, who want to take more active roles in monitoring and associated research and innovation. Our findings reflect UK-based key informant recommendations from a range of professional and locational contexts, as well as references from international case studies in high-income countries. They are based on a relatively small number of national stakeholders (n = 20), purposively sampled for breadth and expertise during this formative design phase, and should therefore be interpreted as a first iteration of user priorities to be complemented by future rounds of engagement at more local scales. As such, we caution that specific infrastructure design priorities may differ significantly in other contexts, especially in low- and middle-income countries or elsewhere where there is less external capacity available for user community-led monitoring, research and innovation activities. This underlines the need to conduct unique user-centred design activities prior to the design and implementation of any new hydrological monitoring infrastructure to tailor services to contextual requirements.

# 4.2 Considerations for Operational Sustainability






Once operational, a mutual realisation of value for infrastructure users and providers improves the infrastructure's sustainability through continued respective contributions. These inputs can generate multiplier effects, whereby contributions towards the infrastructure's growth and improvement increase its value offer, engagement and subsequent contributions over time (Cantor et al., 2021). However, this is contingent on a continuous incorporation of user feedback to keep the value offer relevant and adapted to temporally and spatially evolving user requirements. Channels of feedback should be built into operational services for their periodic evaluation and adaptation (Braud et al., 2020; Cantor et al., 2021).

Given the potential for enabling monitoring infrastructures to grow, and the capacities of their user communities to increase over time, infrastructure providers should consider options for eventual decentralisation of services operation to user community members (Cantor et al., 2021; Widdicks et al., 2024). For the infrastructure provider, this will alleviate the staffing and cost burdens of service provision, whilst for decentralised stakeholders, adopting new responsibilities can improve the quality of local infrastructure services, improve organisational reputations, increase local user engagement and generate similar multiplier effects. The extent to which different infrastructure services can be decentralised, the benefits, and the associated risks of doing so require further research. Subsequently, we now plan to complete more localised and longitudinal user

elicitations for FDRI, as well as catchment-scale pilot projects, to generate evidence and recommendations for the longer-term evolution of its operational structure and governance.

These future developments will also consider how FDRI positions itself within the wider ecosystem of research infrastructures, now that its objectives and major design principles are becoming better defined. Broader initiatives such as eLTER, Horizon and the European Strategy Forum on Research Infrastructures (ESFRI) are examples that offer important opportunities for alignment, particularly through shared protocols, data sharing and interoperability (Ohnemus et al., 2024). While FDRI's initial remit diverges by being more targeted towards hydrological extremes, which addresses a specific monitoring, research and practical challenge in the UK, its design principles resonate with broader international debates on monitoring and research infrastructure design (Nasta et al., 2025; Widdicks et al., 2024). By engaging with similar projects and aligning with broader frameworks where appropriate, FDRI can deliver on its immediate national priorities, while retaining the flexibility to evolve its role and integrate more closely with international research agendas over time towards addressing shared research agendas in the future (Brantley et al., 2017).


## **5 Conclusions**

From multiple methods analysis, we present user recommendations for service delivery in FDRI. We identify 3 key design priorities, which have significant implications for the structuring of equivalent hydrological monitoring infrastructure investments that also seek to optimise user value and outputs from associated research and innovation.



First, prospective infrastructure users broadly recommend that infrastructure providers deliver additional services, where feasible, that are specifically designed to support and enable data collection by their user communities. Cost-effective investments into supporting services for data collection and sharing, such as monitoring site access, telemetry and data hosting services can incentivise data contributions from large user communities, unlocking greater data collection capacities than held by the infrastructure internally. This co-operative approach is also likely to increase the relevance of locally collected data to incentivise closer stakeholder engagement over time. The extent to which decentralised data collection is feasible and cost-effective to support varies according to local contexts. In many cases, its realisation may be a gradual transition while local capacities and incentives to collect data are developed through close engagement with infrastructure user communities.

The second priority is to reserve a part of monitoring infrastructure investments for creating associated communities of users, contributors and innovators, with enabled spaces aimed at facilitating collaborations. Inter-disciplinary collaborations are considered key to genuine state-of-the-art research and innovation, where the sharing of ideas, innovations, opportunities and objectives can lead to the identification of novel research questions and the formation of partnerships to address them. Monitoring infrastructures can catalyse inter-engagements and collaborations in these spaces through enabling support,

including innovation showcase events, investor engagements, intellectual property templates, training workshops, and, in some cases, an active co-ordination of research activities.

Thirdly, user-centred design procedures are now a commonly recommended practice to optimise infrastructure value creation and sustainability. User-centred design ensures that infrastructures are responsive in their services and value offer to stakeholder objectives, their respective activities and their specific requirements for information and support. The procedures implemented in this study should be similarly completed on the catchment scale during infrastructure roll-out to adapt local infrastructure to stakeholder requirements. Periodic evaluations are then needed to ensure that infrastructures remain adaptive and relevant to dynamic user requirements. Infrastructures that remain user-centred and responsive in their design, prioritising value delivery according to the objectives of their stakeholders, in-turn improve their own value proposition by providing better services. By doing so they secure their own sustainability, as the evident benefits of engagement will attract longer term contributions of funding, data, time, personnel, methods, innovations and ideas to sustain and develop them beyond their initial capital investments.

# 5 Data Availability




The interview data is confidential according to ethical and data sharing restrictions. The systematic literature review data is available upon request to the authors.

## **6 Competing Interests**

At least one of the (co-)authors is a member of the editorial board of Hydrology and Earth System Sciences.

# **7 Author Contributions**

William Veness: Writing – review & editing, Writing – original draft, Visualisation, Validation, Software, Resources, Project administration, Methodology, Investigation, Funding acquisition, Formal analysis, Data curation, Conceptualisation.

Wouter Buytaert: Writing – review & editing, Supervision, Methodology, Investigation, Project administration, Conceptualisation.

Alejandro Dussaillant: Writing – review & editing, Supervision, Methodology Investigation, Project administration, Conceptualisation.

Gemma Coxon: Writing - review & editing, Supervision, Validation

Simon De Stercke: Conceptualisation, Methodology

Gareth Old: Writing – review & editing, Administration, Validation

 $Matthew\ Fry:\ Writing-review\ \&\ editing,\ Validation$ 

Jonathan Evans: Writing – review & editing, Validation

#### 650 8 Acknowledgements

We thank all interview respondents for generously giving their time and perspectives. This research was supported by UK Research and Innovation (UKRI) through their funding of FDRI, and the UK Centre for Ecology and Hydrology (UKCEH) as the implementing organisation. Gemma Coxon was supported by a UKRI Future Leaders Fellowship [MR/V022857/1].

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
