# Peer review of "User priorities for hydrological monitoring infrastructures supporting research and innovation"

_EGUsphere, 2025_

## Author Response (AR1)

**Author's Response**

11th September 2025 Editors & Reviewers, *HESS*

Dear Editors & Reviewers,

Thank you once again for your time and carefully considered comments. They have been very helpful in addressing gaps towards a more clearly reasoned and comprehensive manuscript.

We respond here to the comments in turn, indicating where we have made modifications to the article as a result, and/or providing more explanation to defend positions stated in the article.

Best wishes from the Authors.

**Reviewer #1**

This manuscript presents a timely and well-motivated investigation into user requirements for the UK's upcoming Floods and Droughts Research Infrastructure (FDRI), with a broader aim of informing the design of hydrological research infrastructures. The authors combine a systematic literature review with stakeholder interviews, which is methodologically sound and provides a basis for generating practical recommendations. The topic is highly relevant given the ongoing development of environmental monitoring infrastructures and the need for user-driven design.

However, I recommend major revisions for the following reasons:

**1. Clarify the Framing Around Data Scarcity**

The manuscript repeatedly refers to "data scarcity" as a key limitation to hydrological science and innovation (e.g., lines 44–50), yet does not sufficiently engage with the reality that large volumes of hydrological data already exist—including through well-established datasets such as CHIRPS (Funk et al., 2015), the Global Flood Database (Blöschl et al., 2020), CAMELS/CARAVAN (Kratzert et al., 2023), and ISMN. The issue is not simply scarcity, but the cost and complexity of leveraging existing data, including labour, integration, and access barriers. This point is acknowledged in passing (line 49), but it must be integrated more centrally and explicitly into the framing of the manuscript to avoid a misleading narrative.

We agree this can be framed more clearly. We intend to communicate that the issue of scarcity relates in most cases to locally-collected, in-situ or high resolution hydrological datasets and will add that clarification. And indeed for existing datasets (of which there are growing repositories of both remotely-sensed and field-based data as you reference), further issues of data access, sharing and usability remain barriers despite greater availability. We now make this clearer in the introduction.

**Edits:**

Lines 43 – 46: Introduction of data scarcity issue clarified as an issue of shared local data:

"Hydrological science is struggling to address these challenges, and, despite a growing availability of remote sensing datasets, the persistent scarcity of locally-collected, shared data is still cited as a major bottleneck that holds back novel hydrological research and innovation (Chan et al., 2020; FDRI, 2022; Ovink et al., 2023; Paul et al., 2018; Buytaert et al., 2014; Sarni et al., 2018; UN-Water, 2021; Veness et al., 2025)."

Lines 49-55: Modified to acknowledge growing repositories and their role, but their continued challenges of barriers to access/use, and input data scarcity:

"A growing number of global repositories such as the International Soil Moisture Network and the Global Flood Database are pooling remotely sensed and in-situ data and have become valuable resources for advancing hydrological research (Blöschl et al., 2020; Dorigo et al., 2021; Kratzert et al., 2023). Yet these repositories remain constrained both by integration and access barriers, which limit their relevance for decision-making and research at local scales, and by the limited availability of underlying input data. Persistent challenges for collecting and sharing local datasets include the high costs of equipment, installation, and maintenance, as well as

practical difficulties around land access, monitoring security, and data management and dissemination (Addor et al., 2020; Buytaert et al., 2014; Hamel et al., 2020; Paul et al., 2018; Vogl et al., 2017; Veness and Buytaert, 2025)."

**2. Relate More Clearly to Existing Research Infrastructures**

While the manuscript briefly references international infrastructures like TERENO and OZCAR (lines 64–66), it does not go far enough in situating FDRI within the existing ecosystem of RIs, especially eLTER. For instance, Ohnemus et al. (2024) present a comprehensive vision for eLTER RI, which includes a significant hydrological component. It is unclear why FDRI is not part of eLTER, or how it complements or diverges from its goals and structure. Similarly, recent work on pre-implementation (line 133) design of hydrological observatories (e.g. Nasta et al., 2025) and assessments of RI's user needs and surveys on data gaps (Baatz et al., 2018) are highly relevant and must be addressed directly to highlight this study's novelty. The omission of these discussions weakens the positioning of the manuscript.

We agree that the article will benefit from this. We will add to the introduction to provide reader context on how FDRI fits in with these existing domain-specific RIs and observatories, as well as broader research infrastructures like eLTER. Then (relating to your Comment 6), we will offer more reflection in the discussion on the implications of FDRI's design recommendations for its future role and relationships with other infrastructures (in sub-section 'Considerations for Operational Sustainability'). We especially agree that there is a need to relate to those that have broader ecological/environmental focuses such as eLTER in these sections, as the systematic review procedure maintained a narrower focus on hydrological research and data infrastructures.

**Edits:**

Lines 78-84: Lines have been added to the introduction paragraph discussing other RIs and observatories to clarify FDRI's position as suggested:

"Its remit and design are comparable to other national, domain-specific infrastructures, combining intensive data collection at testbed sites with sparser, strategically chosen datasets across wider networks (Nasta et al., 2025; Widdicks et al., 2024). At the same time, FDRI sits alongside broader environmental research infrastructures such as eLTER, an integrated European long-term ecosystem, critical zone, and socio-ecological research infrastructure that incorporates substantial hydrological components (Ohnemus et al., 2024). Although FDRI is not formally embedded within such initiatives, it is intended to complement them by adopting shared design principles and maintaining flexibility for data sharing and interoperability."

Lines 585-594: Discussion now links back to broader infrastructures such as eLTER at the end of the 'Considerations for Operational Sustainability' section:

"These future developments will also consider how FDRI positions itself within the wider ecosystem of research infrastructures. Broader initiatives such as eLTER offer important opportunities for alignment, particularly through shared protocols, data federation and interoperability (Ohnemus et al., 2024). While FDRI's remit is more targeted towards hydrological extremes, its design principles resonate with international debates on research infrastructure design (Nasta et al., 2025; Widdicks et al., 2024). Strengthening such

complementarities can help amplify FDRI's reach and ensure that its national priorities remain connected to European and global research agendas."

**3. Improve Transparency and Justification of Stakeholder Sampling**

The stakeholder interviews form a core pillar of the study, yet the manuscript does not provide enough information to evaluate their representativeness or significance. The authors mention that 20 stakeholders were interviewed and categorized by sector (lines 153–160), but they do not specify the respondents' levels of seniority, expertise, or relevance to RI design and operation. Since the study draws major conclusions from a small sample, this contextual information is critical—especially where only one or two responses appear sufficient to warrant thematic inclusion (Table 1). I recommend the authors clarify the selection process, balance of perspectives, and relative weight given to each respondent type.

We will expand the explanation in Section 2.2 to provide clearer details about respondent selection, including roles, sectors, and levels of seniority. We will also add detail of how preliminary workshops guided the selection of key informants through purposive and snowball sampling of FDRI's major expected user groups. Through FDRI's large project network, we were able to purposively sample key informants from expected major user groups, whilst specifically targeting individuals recognised by others to be knowledgeable about data and research infrastructures like FDRI. Sampling participants with prior experience and understanding of such infrastructures enabled deeper reflections on design elements and priorities.

We emphasise already for transparency in the methods that the sample size (n=20) was intended for breadth and depth in a formative design phase and that further rounds of stakeholder engagement are planned. But on reflection, we can better clarify this in the discussion to ensure the reader is aware of sample size limitations and will add this to the revised manuscript.

**Edits:**

Lines 153-154: We've added some detail as to the scale of the elicitation activities and clear reference to the public resource describing these activities and their findings (FDRI, 2022):

"In 2021, we implemented an initial set of stakeholder consultation activities, including a scoping survey (127 completed), two workshops (81 attendees), and 20 stakeholder group discussions (FDRI, 2022)."

Lines 161-163: Emphasis of minimum 5 years experience and careful selection of knowledgeable respondents has been added:

"Within these groups, we specifically targeted individuals recognised by peers as knowledgeable about hydrological data systems and research infrastructures, with all respondents holding at least five years of relevant experience in their sector. Sampling experienced participants allowed for more substantive reflections on design elements and priorities."

Lines 557-563: In the discussion, the small sample size and limitations for generalisability are now made more explicit and extended:

"Our findings reflect UK-based key informant recommendations from a range of professional and locational contexts, as well as references from international case studies in high-income countries. They are based on a relatively small number of national stakeholders (n = 20), purposively sampled for breadth and expertise during this formative design phase, and should therefore be interpreted as a first iteration of user priorities to be complemented by future rounds of engagement at more local and catchment scales. As such, we caution that infrastructure design priorities may differ significantly in other national contexts, especially in low- and middle-income countries or elsewhere when there is less external capacity available for user community-led monitoring, research and innovation activities."

**4. Refine Terminology and Conceptual Framing**

Several terms used throughout the manuscript are imprecise or informal. For example, referring to "community" as a value category (line 250) feels vague and may trivialize important stakeholder roles. Terms like "expert networks," "cross-sector innovation consortia," or "interdisciplinary research communities" would improve clarity and align better with the discourse on research infrastructure planning. Similarly, "more and better data" (line 252) is too general—more specific terminology regarding data resolution, accessibility, interoperability, or long-term reliability would enhance the precision of the analysis.

Thank you for this observation. The terms used, such as those you highlight, reflect the language commonly used by our interviewees, and we chose to retain them to stay closer to the original expressions of stakeholder perspectives. Also, where they are used, these terms are supported by more specific sub-themes in the tables and are discussed in more detail throughout the text. That said, we agree that the themes can be reworded to better reflect their content, so we have made two changes:

**Edits:**

Table 1: more and better data changed to "data quantity and quality"

Community changed to "user community networks"

Results and conclusions: references to these themes are then updated throughout.

**5. Clarify the Manuscript Structure: Results vs. Discussion**

The manuscript currently presents substantial interpretation and normative claims within the Results section (e.g., section 3.3 on "Structural Design Priorities", or lines 495 onwards). Many of these points—such as design recommendations, innovation pathways, or sustainability implications—would be more appropriately placed in the Discussion section. I recommend reorganizing the manuscript to clearly separate descriptive findings from interpretive insights. Doing so would enhance readability and strengthen the logic of the argument.

Thank you for this point. This raises an important point about what constitutes "results" in interpretative qualitative research, and we see that we can improve the article by adding clarifications and modifications to the manuscript.

Firstly, to address the comment on normative claims, we agree there are many occasions when interviewee perspectives on FDRI are combined with literature references, where language is then used that presents interviewee perspectives more normatively. We will revise phrasing

throughout the results to make it clear when interviewee recommendations made are specific to FDRI. Then, where there is broader support from the literature, that will be noted additionally. This should help distinguish FDRI-specific observations from generalised recommendations in the results.

Secondly, regarding restructuring, we recognise that Section 3.3 of the results in particular reads more interpretive than descriptive, as the priorities presented there are interpretative syntheses based on themes that emerged across the full interview set - not from a single question as in Sections 3.1 and 3.2. We believe this section still belongs in the results. It is common when conducting interviews and inductive analysis for 'emergent' points of emphasis (or themes) such as these to arise that span across different questions. In more structured approaches to qualitative research, these might be omitted from the results. Here we feel that, pragmatically, and given the semi-structured nature of these interviews, the level of emphasis and evidence supporting these structural design priorities warrants their reporting as an important, relevant result from the semi-structured interview and literature review procedure. But we see that an explanation like this is missing currently from the manuscript.

We will therefore address this second point explicitly in the article's final paragraph of the methods. Specifically, this will justify the inclusion of Section 3.3 in the results section and explain how 'design priorities' had a dedicated round of targeted (abductive) thematic analysis inclusive of all questions, in contrast to the thematic analysis Sections 3.1 and 3.2, where only direct responses to single questions were coded. We will also add clearer explanation at the start of Section 3.3 in the results about how these design priorities were recorded separately in the interview analysis.

**Edits:**

Methods - Lines 227-234: Added the previously missing explanation of the abductive coding process used to generate the structural design priorities.

"During the initial rounds of coding, we observed that many interviewee suggestions were not limited to the specific questions asked but also converged around three broad areas of emphasis: services enabling data collection, services enabling community research and innovation, and a need for adaptive infrastructure design. To capture these cross-cutting recommendations, we subsequently organised relevant recommendations from across all interview questions into these abductive thematic coding groups (Saldana, 2021; Saunders et al., 2015). The findings are presented in Sect. 3.3 as "structural design priorities." Although more interpretive than the inductive results in Sects. 3.1 and 3.2, they belong in the Results because they reflect emergent and recurrent points raised independently across multiple interviews, supported by literature evidence, and are directly relevant to the infrastructure design (Galletta, 2013; Saldana, 2021; Saunders et al., 2015)."

Results – Lines 392 – 400: The introduction paragraph for structural design priorities has been updated to better explain how these were derived from the thematic analysis:

"Cross-cutting themes emerged from the interviews that extend beyond question-specific findings, supported by evidence from the literature review. These emphasised the need for hydrological monitoring infrastructures to move beyond traditional models where providers act mainly as data collectors, proprietors and distributors, towards designs that actively engage

and support their user communities. Respondents and literature alike highlighted that such engagement can expand data availability, strengthen research and innovation outcomes, and improve long-term sustainability (Widdicks et al., 2024; Cantor et al., 2021). From this analysis, we identify three structural design priorities for hydrological data infrastructures, which are examined in the subsections that follow."

Results – multiple lines: resolving normative claims where FDRI specific responses were lumped with literature references in their explanation:

Lines 269-274

**Original text:**

By creating a focal point to draw together stakeholders from different industries and research backgrounds, monitoring infrastructures most frequently foster innovation when collaborations form among users with unique combinations of expertise [A1, A4, A6, A8, A9, S1, S2] (Baron et al., 2017; Peek et al., 2020; Fleming et al., 2024; Holzer et al., 2019; Roy et al., 2020; Sartorius et al., 2024; Harrison et al., 2024; Averyt et al., 2018; Widdicks et al., 2024).

**Revised text:**

Respondents highlighted that, by creating a focal point to draw together stakeholders from different industries and research backgrounds, monitoring infrastructures can foster innovation when collaborations form among users with unique combinations of expertise [A1, A4, A6, A8, A9, S1, S2]. This emphasis on cross-sector collaboration is echoed in international experiences, where data infrastructures have been shown to support innovation by convening diverse communities of practice (Baron et al., 2017; Peek et al., 2020; Fleming et al., 2024; Holzer et al., 2019; Roy et al., 2020; Sartorius et al., 2024; Harrison et al., 2024; Averyt et al., 2018; Widdicks et al., 2024).

Line 274: "Two start-up representatives highlight that..." added

Lines 276 – 277: changed to "Four interviewees suggested that partners in such collaborative projects can address weaknesses by filling expertise gaps and cross-validating methods and results [A6, A2, S1, A4].

**Lines 285-293 Original:**

Infrastructures seeking to optimise associated research and innovation should set aside resources for sustaining a community that integrates data users, data providers and major stakeholders in research, innovation and water resources management [T1, A2, A4, R1, A6, A10, C2, I1] (Holzer et al., 2019; Prokopy et al., 2017; Sartorius et al., 2024; Gaillardet et al., 2018; Cantor et al., 2021; Henriksen et al., 2018; Peek et al., 2020; Harrison et al., 2024; Tate et al., 2021; Kiese et al., 2018; Widdicks et al., 2024).

**Revision:**

Eight respondents recommended that FDRI set aside resources to sustain a community integrating data users, providers, and major stakeholders in research, innovation, and water resources management [T1, A2, A4, R1, A6, A10, C2, I1]. The importance of investing in such community-building has also been demonstrated in other infrastructure contexts, where sustained user engagement is critical to long-term scientific and operational impact (Holzer et

al., 2019; Prokopy et al., 2017; Sartorius et al., 2024; Gaillardet et al., 2018; Cantor et al., 2021; Henriksen et al., 2018; Peek et al., 2020; Harrison et al., 2024; Tate et al., 2021; Kiese et al., 2018; Widdicks et al., 2024).

Line 304: "three respondents suggested that" added

Line 337: "flagged by four respondents" added

Line 338: "interviewees also recommended" added

Line 345: "among respondents" added

Line 346: "citizen scientists explain that" added

Line 352: changed to separate interview evidence from literature: "Interviewees emphasised that such a platform should be openly accessible and, where feasible, provide real-time and visualised data that is navigable by the public while remaining useful for expert users via APIs and download options [S1, S2, I1, A10], aligning with user-centric platform features reported elsewhere (Dallo & Marti, 2021; Jones et al., 2015)."

Line 355: "Cantor et al. (2021) and Widdicks et al. (2024) recommend..." added

Line 360: "respondents highlight the need for" added

Line 362: "interviewees recommend that" added

Line 375: "suggested by respondents" added

Line 383: "Four respondents with backgrounds studying or actively managing hydrological hazards explain the benefit of" added

Line 395: "two academics warn that" added

Line 424: "respondents widely made recommendations to" added

Line 431: "which is a view shared by recent studies" added

Line 437: "are expected to" added

Line 459: "recommended priority" added

Line 462: "will" changed to "expected to"

Line 463: "may" added to reflect uncertainty of the suggestion

Line 479: "should" added

Line 481: "can" added

Line 494: "by the regulator representative" added

Line 499: "recommended as an additional service by thirteen respondents" added

Line 519: "among respondents" added

**6. Strengthen the Discussion and Critical Reflection**

The Discussion section (Section 4) currently functions more as a continuation of the results, reiterating conceptual points rather than reflecting critically on the implications of the findings. A deeper discussion is needed on how FDRI can position itself within national RIs, related RIs in other countries, and or internationally, drawing on similar concepts such as transnational access frameworks (e.g., ESFRI, Horizon Europe), and how its structure might evolve in light of lessons learned from other RIs. Engaging more directly with European strategies for open data, FAIR principles, and transdisciplinary collaboration would help clarify what makes FDRI unique and where it might integrate or diverge from existing models.

We agree that situating FDRI within the wider ecosystem of national and international research infrastructures is important, and we have expanded Section 4.2 ("Considerations for Operational Sustainability") to reflect this. In the revised manuscript, we explicitly reference eLTER, Horizon Europe and ESFRI, as well as principles of interoperability, data sharing, and open frameworks. We also note how FDRI's remit diverges from these broader initiatives by focusing more narrowly on hydrological extremes in the UK, addressing a significant national research and monitoring gap.

At this stage of development, however, our priority is to convey the value of establishing a sustainable UK niche that avoids redundancy and delivers services of immediate benefit to national stakeholders. Therefore, whilst it is important to show awareness and general alignment with these initiatives as we have now added, we have decided not to extend the discussion too much here, which we feel would be premature given FDRI's early design stage. Instead, we emphasise the importance of maintaining flexibility so that, once its core national role is established, FDRI can start to engage more proactively with broader international initiatives towards the 2029 operational phase.

**Edits:**

**Lines 585-594:**

"These future developments will also consider how FDRI positions itself within the wider ecosystem of research infrastructures, now that its objectives and major design principles are becoming better defined. Broader initiatives such as eLTER, Horizon and the European Strategy Forum on Research Infrastructures (ESFRI) are examples that offer important opportunities for alignment, particularly through shared protocols, data sharing and interoperability (Ohnemus et al., 2024). While FDRI's initial remit diverges by being more targeted towards hydrological extremes, which addresses a specific monitoring, research and practical challenge in the UK, its design principles resonate with broader international debates on monitoring and research infrastructure design (Nasta et al., 2025; Widdicks et al., 2024). By engaging with similar projects and aligning with broader frameworks where appropriate, FDRI can deliver on its immediate national priorities, while retaining the flexibility to evolve its role and integrate more closely with international research agendas over time towards addressing shared research agendas in the future (Brantley et al., 2017)."

**Conclusion**

This manuscript addresses a critical and contemporary issue and makes a valuable contribution in concept. However, a more precise framing, fuller engagement with related

work, clearer stakeholder justification, and improved structural organization are needed to realize its full potential.

Baatz et al. 2018 https://doi.org/10.5194/esd-9-593-2018

Blöschl et al. 2020 https://doi.org/10.1038/s41586-020-2478-3

Funk et al. 2015 https://doi.org/10.1038/sdata.2015.66

Kratzert et al. 2023 https://doi.org/10.1038/s41597-023-01975-w

Nasta et al. 2025 https://doi.org/10.5194/hess-29-465-2025

Ohnemus et al. 2024 https://doi.org/10.1016/j.indic.2024.100456

Thank you and best wishes from the authors.

**Reviewer #2**

This manuscript describes a research project seeking to inform an effort to improve hydrological monitoring infrastructure through deliberate user engagement about priorities. The manuscript is clearly written, the methods are sound, the contribution is laid out in a logical and clear fashion, and the manuscript does what it sets out to do. It makes a nice contribution to the field of user-centered water data system development.

I just have a very few minor edits to suggest.

First, in the last paragraph of the introduction (85-90) it would be helpful to add one or two sentences on what questions the research seeks to answer- e.g., "This study seeks to learn more about the specific end-user data needs and priorities to inform development of the FDRI so that the system is responsive and useful" or something along those lines.

Second, in Figure 3, it may be helpful to draw a line between "cost" and "supporting infrastructure" as these seem related.

Third, in the references, there are a few places where an additional paragraph break/ line space is needed where multiple references run together.

This is a clearly written paper that makes a nice contribution to a growing body of scholarship on user-driven data systems and I look forward to its publication.

We thank the reviewer for taking the time to provide thoughtful feedback on our manuscript and for recognising its contribution to user-centred hydrological monitoring system development.

We appreciate the helpful suggestions for improvement. For the first point, we agree this is important. We will add one to two sentences to the final paragraph of the introduction to clarify the specific research questions the study seeks to address. The second observation is also very helpful and highlights a link that didn't make it into the visualisation - we will update Figure 3 to include a connecting line between "cost" and "supporting infrastructure" to better illustrate this relationship. We will also carefully check the reference list and ensure paragraph breaks are correctly formatted.

**Edits:**

Lines 96-98: Objectives of the study are added to the final paragraph of the introduction that outlines the document:

"In this study, we identify end-user needs and priorities in the context of the FDRI investment. Specifically, we aim to establish what data and service needs are most important to potential users, how these shape design priorities for FDRI, and what implications they carry for the evolution of hydrological monitoring and research infrastructures more broadly."

Figure 3 updated with connecting line.

References fixed and formatted.

Best wishes from the authors.

**Reviewer #3**

Thank you to the reviewer for their detailed comments that will certainly improve the manuscript. And we agree, the findings on community elements are interesting and should generate interesting lines of research and scientific debate. We reply to each of the detailed comments in-turn:

**General comments**

The paper presents a study aiming at defining the needs and priorities of future users of the Floods and Droughts Research Infrastructures, which is currently developed in the UK.

The study is based on literature review and on semi-structured interviews of future users. The paper is generally well written and clear. It is informative and provides results that are somehow unexpected (like the majority or responses highlighting the "community" theme (line 251).

The paper is UK centered and could benefit from a better positioning within what is done in the international community (see some suggestions below).

I suggest moderate revision of the paper to address the specific points listed below.

**Specific comments**

1/ A general comment: in the manuscript, the use of the word "infrastructure" is ambiguous, as, to my understanding, it is use for things that are different. The word can refer to physical elements in the fields including sensors, a data information system or an organization that provides services, like its use in the context of European Strategic Forum for Research Infrastructures (ESFRI), in particular the eLTER RI (Integrated European Long-Term Ecosystem, critical zone and socio-ecological Research, https://elter-ri.eu/). This ambiguity makes the reading of the paper sometimes confusing. Clarification of the meaning or the use of other words within the paper is recommended.

We thank the reviewer for highlighting this important point. We agree that our initial use of the term "infrastructure" was at times ambiguous, as it referred both to broader research infrastructures (such as eLTER) and to the physical and digital elements that comprise FDRI. To address this, we have made several changes in the revised manuscript:

Introduction: we now provide a clearer description of the infrastructure landscape, extending explanation of how FDRI is positioned relative to other national and international RIs (lines 67-84).

Throughout results: where we describe the fixed, mobile, and digital parts of FDRI, we now refer to these as components/services rather than "infrastructure," to avoid confusion.

2/ Line 44: the argument about data scarcity in hydrological sciences is not necessarily true now, with the avenue of satellite data products with a spatial and temporal resolution which becomes relevant for hydrological studies and provide high resolution data that can be useful in water resources management and in running or evaluating models.

We thank the reviewer for raising this important point, which echoes Reviewer 1's related comment on the framing of "data scarcity." We have revised the introduction to clarify that current challenges relate less to an absolute scarcity of data and more to the availability and

usability of locally collected, in-situ, or high-resolution datasets. We now explicitly note that while there are extensive satellite and global datasets available, these often remain constrained by barriers of access, integration, and relevance at local scales.

**Edits:**

Lines 43 – 46: Introduction of data scarcity issue clarified as an issue of shared local data:

"Hydrological science is struggling to address these challenges, and, despite a growing availability of remote sensing datasets, the persistent scarcity of locally-collected, shared data is still cited as a major bottleneck that holds back novel hydrological research and innovation (Chan et al., 2020; FDRI, 2022; Ovink et al., 2023; Paul et al., 2018; Buytaert et al., 2014; Sarni et al., 2018; UN-Water, 2021; Veness et al., 2025)."

Lines 49-55: Modified to acknowledge growing repositories and their role, but their continued challenges of barriers to access/use, and input data scarcity:

"A growing number of global repositories such as the International Soil Moisture Network and the Global Flood Database are pooling remotely sensed and in-situ data and have become valuable resources for advancing hydrological research (Blöschl et al., 2020; Dorigo et al., 2021; Kratzert et al., 2023). Yet these repositories remain constrained both by integration and access barriers, which limit their relevance for decision-making and research at local scales, and by the limited availability of underlying input data. Persistent challenges for collecting and sharing local datasets include the high costs of equipment, installation, and maintenance, as well as practical difficulties around land access, monitoring security, and data management and dissemination (Addor et al., 2020; Buytaert et al., 2014; Hamel et al., 2020; Paul et al., 2018; Vogl et al., 2017; Veness and Buytaert, 2025)."

**3/ Line 69: explicit the acronym WRM**

Thanks, changed to 'water management' as the acronym was not re-used

4/ Lines102: is the Water4All European partnership relevant for the UK (https://www.water4all-partnership.eu/). This project is also conducting a survey to define the users' needs.

This is indeed relevant, thank you. We have included this in the introduction (Line 82) when discussing the wider landscape of infrastructures and initiatives.

**5/ Lines 119, 125: ensures and note ensure**

Thanks, modified.

**6/ section 2.1: I am surprised not to see the words "information system" in the search query.**

We agree this search term would have been appropriate and should have been included. However, we expect that the inclusion of the terms "monitoring", "observatory", "hydrometry" and "data" has captured most or all of the literature containing the phrase 'information systems' also within their article.

7/ line 147: if I understood correctly, you launched the project using online questionnaire and online workshops. Could you have a bias in the participation of such events, that tends to attract persons that usually participate in such events. How do you ensure that you have reached all the potential users?

We thank the reviewer for this important point. We have expanded the Methods section to provide more detail on the scoping activities that preceded the interviews. These included an online questionnaire completed by 127 respondents, two workshops with a total of 81 participants, and 20 stakeholder group discussions. These activities were designed to be inclusive and representative of FDRI's expected user groups, and they informed the iterative identification of priority users, services, and objectives for the infrastructure. Any new potential user groups identified during these activities would be discussed internally and invited to future stakeholder elicitation activities. While we recognise that events may carry some self-selection bias, the diversity of participation across sectors (civil society groups, public, academics, industry, regulators, government, private sector, startups) helped mitigate this. To ensure transparency, we have added a citation to the full FDRI Community Report (FDRI, 2022), which provides further detail on the scoping process (lines 153-154).

(https://www.ceh.ac.uk/sites/default/files/2022-06/FDRI\_Community%20Report\_FINAL.pdf).

**8/ Line 212: could you explain what an inductive approach is?**

We thank the reviewer for highlighting this. We have added a short clarification in the Methods to explain what we mean by an inductive approach:

Lines 220-221: "whereby codes and themes are not pre-set in advance, but instead emerge from the data through the analyst's interpretation of participant responses"

In brief, this refers to a qualitative research method where codes and themes are generated directly from the data, rather than being pre-determined, allowing the analysis to be guided by participant responses.

9/ Line 335: users ask for real-time access to data. This objective is more an operational one and is generally the role of monitoring networks. However, the initial objective of the FDRI network is "to support state-of-the-art research and innovation" (line 72). This seems somehow incompatible.

Thanks, this touches upon a prudent debate on real-time data that we can elaborate more clearly in the text. One of FDRI's priorities is to support practical research to support better management of floods, droughts and other hydrological issues in the UK. As such, studies such as those involving citizen scientists (e.g. surge monitoring in 24 hours after a flood), and studies evaluating operational or management activities sometimes require real-time data to trigger or inform subsequent data collection efforts. It is also an important additional incentive/reward for engagement with research activities for civil society, community and industrial data user/contributors if their efforts yield real-time data, insights or feedback that is interesting or useful to them. On the other side of the debate, real-time data is more expensive and comes with operational risks such as telemetry failure, so this is a decision to take case-by-case considering the benefit, costs and operational risks of the data being real-time.

Edits:

Lines 394-404: We have updated the paragraph discussing real-time data:

"Despite these potential benefits, two potential users caution that providing real-time data access can create operational reliance on the data, with high expectations of platform uptime and performance [C2, N1]. Two academics warn that this may also go against the core mission of infrastructures like FDRI if they are primarily intended to support research and innovation rather than replacing operational infrastructure such as flood information systems [A7, A9]. Investing in ultra-reliable real-time services for operational systems may divert resources from core research and innovation functions that rely less on immediate data accessibility [A7, A9]. Nonetheless, there are many opportunities for aggregated monitoring infrastructures to provide new insights, validation and other data services for operational systems [A2, A7]. Furthermore, real-time data can support novel practical research applications such as rapid post-event studies and citizen science campaigns, whilst providing additional incentives for user contributions if data can be immediately viewed. Hence, fulfilling these opportunities whilst managing expectations and averting misuse in risk contexts requires planning and potential partnership with other data services acting in the public interest (Collins et al., 2016; Dallo and Marti, 2021; Stephenson et al., 2018)."

10/ Lin 355: FAIR principles are only principles and their practical implementation may lead to different interpretations. Their meaning within the context of the FDRI program should be defined.

Thank you for highlighting this. We highlight some key priorities raised by prospective users in the manuscript (e.g. Lines 362-372 discussing DOIs, reproducibility repositories, metadata).

But we have now also added that the precise implementation within FDRI will be determined in subsequent design stages. We note that this will involve further elicitation, comparison with equivalent infrastructures, and testing with early adopters and users to identify what works most effectively in practice:

Lines 375-377: "The detailed implementation of these standards within FDRI will be defined in later design stages through engagement with equivalent infrastructures and testing with early adopters."

11/ Open Geospatial Consortium (OGC) standards are not only relevant for remote sensing date. They are also relevant for vector data.

Thank you, we have added "and vector" to Line 373.

12/ p13: I am surprised not to see the harmonization of protocols to measure the targeted observed properties as an issue. It proves to be a challenge (for instance, in the eLTER RI, it required several years of work (Zacharias et al, 2025)

Thank you, this is a very good point and it is brilliant to see this research by Zacharias et al. (2025). This was raised by respondents T1, A6 and A9 in their responses, but was coded to other labels such as standardisation, FAIR principles, quality assurance and control and interoperability. We agree this is a major issue and should be addressed explicitly in the text. Harmonisation of protocols has been added to Section 3.2.2 in digital products and services:

Lines 378-381: "For integrating external data contributions, respondents also emphasised the importance of harmonising measurement protocols to ensure comparability across sites and contributors [T1, A6, A9], for which the eLTER research infrastructure recently defined an adoptable Framework of Standard Observations (Zacharias et al., 2025)."

**13/ Line 364: "automated by manually verified processes": could you elaborate on the way you see the implementation of such processes?**

Thank you – we have extended the description of this in the text to make it clearer:

Line 385: "automated data quality control that is manually verified following anomaly alerts and during periodic audits"

**14/ Line 371: I am surprised to see that the used of X or Facebook is suggested. With such platforms, how do you ensure the protection of personal data?**

We will add a note on this, but this is largely just as a data, insights and research communication & tool where data sharing agreements allow. We have now added a note where this is mentioned on compliance with data sharing agreements and personal data protection requirements:

Line 389: "These platforms, particularly X (formerly known as Twitter) and Facebook (Stephenson et al., 2018), are used regularly by researchers and practitioners as well as the public, and they are an under-utilised medium of communication, awareness-raising and coordination [N2] (Stephenson et al., 2018). Any such use of data on these platforms would need to comply with data-sharing agreements and personal data protection requirements."

**15/ p.15: would the paper of Brantley et al. (2017) be relevant for your discussion?**

Thank you and yes, this is very relevant. We have included the reference in the introduction when introducing critical zone observatories (Line 67), and also in the discussion to support the case the FDRI should explore wider integration in the future towards addressing broader, shared research agendas (Line 594).

**16/ Figure 3: could you add some elements in the caption on how to read this figure?**

Yes, thank you for flagging this. New caption:

"Figure 3: Thematic summary of perceived barriers to field testing of innovations in response to Q21: We will be using sites as innovation testbeds... What are the current barriers to field testing of innovative technologies? The symbols scale to the number of interview references made to each code (light blue) and theme (dark blue), and dashed lines represent overlap between themes."

**17/ lines 582-591: this paragraph is more discussion than conclusion**

Agreed. This whole paragraph fits much better at the end of section 4.1 where it now sits (Lines 554-565). Thank you for your review and thoughtful comments.